# Estimation of Prevalence of Osteoporosis Using OSTA and Its Correlation with Sociodemographic Factors, Disability and Comorbidities

**DOI:** 10.3390/ijerph16132338

**Published:** 2019-07-02

**Authors:** Peizhi Wang, Edimansyah Abdin, Saleha Shafie, Siow Ann Chong, Janhavi Ajit Vaingankar, Mythily Subramaniam

**Affiliations:** Research Division, Institute of Mental Health, Singapore 539747, Singapore

**Keywords:** osteoporosis, Osteoporosis Self-Assessment Tool for Asians, disability, comorbidities

## Abstract

Osteoporosis is a growing concern for an aging society. The study aimed to estimate the prevalence of older adults who were at risk of osteoporosis and explore factors associated with osteoporosis. The relationship between the risk of osteoporosis, chronic conditions and disability was also explored. We hypothesized that respondents with high risk index of osteoporosis would be associated with greater disability. Participants aged 60 years and above (*N* = 2565) who were representative of Singapore’s multiethnic population were recruited. The Osteoporosis Self-Assessment Tool for Asians (OSTA) was used to classify the risk of osteoporosis. Information on sociodemographic details and chronic diseases were collected, while severity of disability was measured using the World Health Organization Disability Assessment Schedule 2.0. The overall prevalence of the respondents who were at risk of osteoporosis was 52%. Those belonging to an older age, Chinese, female, never married or widowed, lower education and retired were associated with a higher risk of osteoporosis. A diagnosis of diabetes or hypertension was a protective factor against the risk of osteoporosis. High risk of osteoporosis was not associated with disability. Our findings highlighted specific factors associated with the risk of osteoporosis that could be useful for the prevention of osteoporosis and fractures.

## 1. Introduction

Throughout the world, the proportion of elderly people has been steadily increasing over the last decade [1]. The World Health Organization reported that people aged 60 years and older make up over 11% of the global population [2]. By 2050, the number is projected to rise to about 22% [2]. A nationwide study conducted in Denmark has suggested that the estimated prevalence is the tip of the iceberg [3]. Based on their analysis, osteoporosis is significantly under diagnosed and under treated, which is probably the same as elsewhere in the world [3]. This statement is worrying for countries that have an aging population, especially Singapore, which has one of the world’s fastest ageing populations [4].

As one ages, several physiological and nutritional changes occur. It has been frequently reported that older people have reduced food intake and lack variety in their diet, which makes them vulnerable to malnutrition and micronutrient deficiency [5]. On the other hand, with increasing age; height, weight, muscle mass and bone mass decreases [6]. These factors predispose the elderly towards frailty and increase the risk of osteoporosis [7].

Osteoporosis is a disease characterized by reduced bone mass and deterioration of microarchitecture of bone tissue leading to increased bone fragility and fracture risk [8]. In Singapore, hip fractures are projected to increase from 1300 in 1998 to 9000 in 2050 [9]. Among the Asian countries, Singapore’s age-adjusted hip fracture rates for females above 50 years are currently among the highest in the world [9]. Hip fractures are also associated with increased mortality and morbidity due to prolonged immobilization [10]. A report has projected that the cost attributed to osteoporosis related hip fracture is USD 17 million [9]. Besides the financial cost for the treatment of hip fractures, social dependency is also high. Individuals that sustain osteoporotic hip fractures have a mortality rate of 20% to 27% at year one [10]. Among those who survive, months of rehabilitation are required to become ambulatory. Therefore, it would be important to diagnose elderly who are osteoporotic and start treatment early to delay further loss of bone mass or to increase bone density. The current technology to assess bone mineral density (BMD) is through dual-energy X-ray absorptiometry (DXA). Generally, low bone density can be categorised into two groups according to the WHO criteria. BMD that lies between 1 standard deviation and 2.5 standard deviation below mean can be categorised as Osteopenia [11], while values that are 2.5 standard deviations and more below the mean are categorised as osteoporosis [11]. While DXA is widely used and deemed as the gold standard, it is not recommended for mass screening due to its related cost [12,13]. Thus, a simpler tool would be more useful to identify individuals who are at risk of osteoporosis. The Osteoporosis Self-Assessment Tool for Asians (OSTA) was developed and validated in eight Asian countries [14]. While the OSTA was originally designed for screening Asian females, it was also reported to be effective for males using different cut-offs [15] The categories are primarily based on its association with BMD. This measure can provide an indirect estimate of the prevalence of older adults who are at risk of osteoporosis.

Osteoporosis has also been linked to other medical conditions, poor health, reduced functioning and poor social outcomes. For example, osteoporosis has been linked to cardiovascular disease, sarcopenia and rheumatoid arthritis [7,16,17]. It has been suggested that reduced BMD is linked to lower quality of life [18,19,20]. However, what is surprising is that the review article by Wilson (2012) reported that physical aspects of the health-related quality of life are worse in those with low BMD even without fractures [20]. Having said so, the review recognised that it was inconclusive as the methodology for classifying other fractures was not adequately described in the original articles [20].

The possibility that quality of life may be affected has been postulated due to comorbidities [18]. A recent study by Cauley and colleagues demonstrated that females with comorbidities (i.e., depression) have a higher risk of fracture [21]. These comorbidities may exist in patients with osteoporosis, even before a fracture occurs and as a result impair their activities of daily living.

The main aim of this study was to estimate the prevalence of osteoporosis based on OSTA. In addition, the study also intended to explore the (i) sociodemographic factors, which can predict OSTA in the elderly and finally, (ii) relationship of OSTA and disability in the elderly population.

## 2. Materials and Methods

### 2.1. Sample

The data for the current study was extracted from Well-being of the Singapore Elderly (WiSE). This was a comprehensive single phase, cross-sectional population-based epidemiological survey. Participants in the study were Singapore residents (including Singapore Citizens and Permanent Residents) aged 60 years and above who were living in Singapore at the time of the survey. This study also included respondents who were in day care centres, nursing homes and institutions. Respondents were randomly selected from an administrative database of residents in Singapore. A total of 2565 respondents were recruited and disproportionate stratified random sampling was used to ensure the appropriate and equivalent proportions of the three main ethnic groups (i.e., Chinese, Malay and Indian). Individuals aged ≥75 years old were over sampled. The study methodology has been described previously in greater detail [22].

Written informed consent was obtained from respondents. In cases where respondents were unable to provide consent, consent was taken from their legally acceptable representative/next of kin. The study was approved by the relevant ethics committee: National Healthcare Group Domain Specific Review Board (DSRB) and the SingHealth Centralised Institutional Review Board (CIRB).

### 2.2. Main Instruments

#### 2.2.1. Sociodemographic and Clinical Questionnaire

Sociodemographic information was recorded in a standardized data collection form. This included questions on age, gender, ethnicity, marital status, education and employment status. Participants were asked if they had been diagnosed with any of the following chronic illness: hypertension, heart problems, stroke, diabetes, transient ischemic attack, or depression. The study used the 10/66 algorithm to diagnose dementia [23]. Weight was measured with respondents wearing indoor clothing without shoes, using a digital standing scale.

#### 2.2.2. The Osteoporosis Self-Assessment Tool for Asians (OSTA)

OSTA is an index for classifying the risk of osteoporosis among Asians. The index is based only on age and body weight, utilizing the formula of 0.2 × [body weight (kg) − age (years)]. Participants were classified into three categories accordingly to their gender. We defined “at risk” as adults who either had an intermediate or high-risk index.

For postmenopausal females, a cut-off point of −1 is used, which yields a sensitivity of 91% and specificity of 45% compared to DXA. The index comprises three categories: (i) Low Risk Index > −1 (ii) Intermediate Risk Index −1 to −4 and (iii) High Risk Index < −4 [14].

For Asian men (50–90 years old), a cut-off point of −1 is similarly recommended. OSTA value of ≤ −1 has a sensitivity of 81% and specificity of 66% compared to DXA. The index was validated with a separate sample of 356 men with a sensitivity of 83% and specificity of 67%. The OSTA for men comprises three categories: (i) Low Risk > −1 (ii) intermediate Risk Index −1 to −6 and (iii) High Risk Index < −6 [15].

#### 2.2.3. World Health Organization Disability Assessment Schedule 2.0 (WHO-DAS 2.0)

Disability was assessed using a 12-item WHO-DAS 2.0 questionnaire, which was administered to all elderly or their caregiver if the elderly were unable to answer the questions [24]. This scale is used to assess functioning in six domains: Cognition (understanding and communicating); Mobility (moving and getting around); Self-care (hygiene, dressing, eating and staying alone); Getting along with others (interacting with other people), Life Activities (domestic responsibility, leisure, work and school); and Participation (joining in community activities). The 12-item WHO-DAS 2.0 provides a reliable and valid measure of disability. Respondents were asked to recall how much their disability interfered with their lives in the last 30 days on a five-point response scale from 0 (none) to 4 (extreme/cannot do). The responses were summed and weighted then all six weighted scores were converted into a summary score ranging from 0 to 100, with higher scores reflecting greater disability.

## 3. Statistical Analysis

All estimates were weighted to adjust for oversampling and post-stratification sampling for age and ethnic distributions between the sample and Singapore’s resident population. Weighted mean and standard error were calculated for continuous variables, and weighted percentages and standard errors (SE) for categorical variables. To examine the associations between OSTA categories and sociodemographic variables, Chi-square (*χ*^2^) tests were used in the bivariate analysis followed by multiple logistic regressions in multivariate analysis. Age group, ethnicity, gender, marital status, education and employment status were included as sociodemographic correlates of OSTA. To examine the association between clinical conditions and WHO-DAS 2.0, multiple logistic regression analysis was used while adjusting for sociodemographic factors. SE and significance tests were estimated using the Taylor series linearization method. All statistical analysis was carried out using Statistical Analysis Software (SAS) System version 9.2 (SAS Institute Inc., Cary, NC, USA). Statistical significance was evaluated at the <0.05 level using two-sided tests.

## 4. Results

In all, 2565 face-to-face interviews were completed, yielding a response rate of 65.6%. Due to missing data, OSTA could only be calculated for 2345 (44.7% male and 55.3% female) respondents. The mean (SE) age of the sample was 69.9 (0.1) and ranged from 60 to 105 years.

Table 1 presents the weighted proportions of OSTA category amongst the sociodemographic variables in the overall sample. The overall prevalence of the respondents who were at risk of osteoporosis was 52%. Out of these, 37.1% among the males and 64% among the females were at risk of osteoporosis. Among the females, 24.2% were in the low-risk category, 52.9% in the intermediate risk and 23% in the high-risk category. For the males, 51.7% were in the low-risk, 44.8% on the intermediate risk and 3.5% were in the high-risk category.

Table 2 shows the sociodemographic correlates of osteoporosis. Multiple logistic regression analysis revealed that respondents aged 75–84 years old were more likely (OR = 15.6) to be at risk of osteoporosis than those aged 60–74 years. For those aged 85 years and above, the OR was not estimated due to low sample size. Among the ethnic groups, non-Chinese were less likely to be at risk of osteoporosis. Those who were at risk of osteoporosis were more likely to be females, never married or widowed, did not complete their secondary education, and retired. The *r*^2^ of the regression model was 41.8%. These results were summarized in Table 2.

The association of health conditions with risk of osteoporosis were examined using logistic regression analyses, and are reported in Table 3. Respondents who have hypertension (OR = 0.6) or diabetes (OR = 0.6) were less likely to be at risk of osteoporosis.

Individuals who were at intermediate and high risk of osteoporosis had a significantly higher WHO-DAS 2.0 score [M = 11.24, S.E. = 0.58, 95% CI (10.10, 12.40)] compared to those who were at low risk [M = 5.0, S.E. = 0.43, 95% CI (4.19, 5.89)]. However, this difference was not significant after adjusting for sociodemographic correlates [*β* = 0.35, 95% CI (−1.16, 1.86)]. We further analyzed the OSTA severity categories with WHO-DAS 2.0. Our results revealed that there was no association between OSTA severity with WHO-DAS2.0 functioning. Compared to the low risk group, intermediate [*β* = −0.93, 95% CI (−2.41, 0.56)] and high-risk group [*β* = 2.46, 95% CI (−1.06, −6.00)] did not have higher WHO-DAS 2.0 scores.

## 5. Discussion

This study found that the prevalence of elderly at risk of osteoporosis was 52% in Singapore. This prevalence figure translates to about 278,000 older adults who are at risk of osteoporosis in Singapore in the year 2011. Of these, 76,000 older adults were in the high-risk category. Although no nationwide studies utilised OSTA to estimate the prevalence of osteoporosis, a US report has estimated that there are 37–50% of all female and 28–47% of all men aged 50 years and above have osteopenia [25]. Technically, OSTA categories are based on the risk estimates for osteoporosis. However, we compared our prevalence against osteopenia as this condition is characterised by low BMD and is less severe than osteoporosis. The OSTA scale has good sensitivity, however the specificity of the scale is moderate compared to DXA. Thus, the risk score could be used conservatively to identify individuals who are likely to have low BMD. Having said so, this scale is not without its merit. It has allowed us to potentially estimate the number of people with risk of osteoporosis which will be essential for the planning of services and initiatives. Our results indicate that females were at higher risk. This is not surprising as studies have consistently reported higher incidence of fracture among the females [26,27,28,29]. Physiologically, females have smaller bones and tend to lose bone mass more quickly after changes to their hormones during menopause. However, despite the lower prevalence of osteoporosis in males, the associated mortality in males is higher compared to females [29].

Compared to Chinese, non-Chinese (Malays, Indians and Others) have a lower risk of osteoporosis. This result is consistent with the previous study looking at the crude incidence rate in Singapore [27]. Moreover, OSTA score is based on age and weight of an individual. Referring to Singapore National Health Survey, Malays and Indians are more likely to be overweight and obese, which is a protective factor for osteoporosis [29,30]. Differences in bone mineral density due to race/ethnicity has been reported elsewhere [31,32]; African American and Asian men have thicker cortices and higher trabecular volumetric BMD compared to the Whites, which may increase bone strength [32].

Individuals who were never married or widowed were at a higher risk of osteoporosis. This result is consistent with studies, whereby there is a significant higher risk of fracture among unmarried and widowed females [33,34]. A systematic review concluded that there was a protective effect of being married relative to the risk of osteoporotic fracture [35]. It has been suggested that the beneficial effect of marriage maybe due to better nutritional status and healthier lifestyle [36]. More recently, a study suggested that psychosocial stressors (e.g., being widowed) have an impact on bone health [36]. Having said that, the effects of bereavement on one’s bone health has not been well studied. Thus, we can only speculate that grief and/or other external factors such as drop in income, changes in lifestyle, habits and etc. would have affected bone health leading to the association between being widowed and risk of osteoporosis.

Previous studies have suggested socio-economic factors are predictors of osteoporosis [34]. Socioeconomic factors such as education and employment status are related to variations in behaviour. We found that lower education and retirees were more likely to be at risk of osteoporosis. Education has been hypothesized to enable people to integrate healthy behaviours (e.g., dietary choices, nutrition, access to healthy food, exercise) into their lifestyle, which gives them a sense of control over their health [37]. An individual with higher education is also more likely to take supplements (e.g., Vitamin D, calcium tablets) or be on hormone replacement therapy [38]. Currently, hormone replacement therapy is an important intervention strategy for the prevention of bone loss in post-menopausal females. Retirees on the other hand may become inactive as they stop working. Consequently, retirees would be at a higher risk for osteoporosis. In fact, physical activity has been reported to positively influence bone density [39,40,41]. In Asian culture, the Chinese ideal of retirement is to *xiang qing fu* which translated literally means “enjoying the fortune of doing nothing” [42]. Such traditional concepts, may lead to physical activity not being considered as a priority for the older adults after retirement leading to a higher risk of osteoporosis.

Apart from sociodemographics, we also examined the chronic medical conditions of the older adults. We found that the presence of diabetes was associated with a decreased risk of osteoporosis. In Singapore, majority of diabetic patients are diagnosed with type 2 diabetes [43]. Because type 2 diabetics are associated with being overweight, bone density may be increased. However, studies looking at the association between weight and BMD are conflicting. Greco and colleagues reported that overweight was neutral or protective for BMD [44]. Whereas, obesity was associated with low BMD [45]. Some studies also suggest that while diabetes is associated with normal or even high BMD, there may be a reduction of bone strength, that is, not reflected in the measurement of BMD [46].

We also found a lower risk of osteoporosis for respondents who were diagnosed with hypertension which is contrary to other studies that have identified a higher risk of osteoporosis for individuals with hypertension [47,48]. To control blood pressure, individuals have often been advised to make lifestyle changes such as reduction in salt intake and exercise. This reduction of salt intake may have a positive effect of bone health, as suggested by Carbone [49]. Furthermore, commonly prescribed antihypertensive medication such as thiazide diuretics, spironolactone, beta blockers, angiotensin-converting enzyme inhibitors and nitrates have been suggested to have a positive effect on BMD [50]. Further research needs to be done to understand this phenomenon completely.

Increased risk of osteoporosis was not associated with disability. Osteoporosis may also be regarded as a silent epidemic, whereby individuals might not be aware of their condition. In other words, individuals with low bone density are able to proceed with their life as per normal and are able to perform everyday tasks; thereby do not experience any impact on disability until they sustain bone fractures. However, this study did not establish a history of bone fractures among the respondents.

The findings of the study should be interpreted in view of some limitations. First, BMD was not measured. Risk of osteoporosis was based on the formula of OSTA, which is validated up to 88 years in females and 90 years in males. However, our data comprises individuals who are above this age range. Secondly, our response rate was 65.6% and not all participants agreed for their weight to be measured. It is possible that those who refused to participate in our study or measure their weight belong to a more vulnerable population (e.g., physically disabled). Taking this into consideration, the prevalence could be higher. Thirdly, we did not measure the estrogen levels of the individuals, which may affect the bone health of the females, which could act as a confounder. Lastly, the cross-sectional design of the study also did not permit us to determine any causal relationships. In view of the high prevalence rate of risk of osteoporosis based on the OSTA, we are cautious of the results.

## 6. Conclusions

This study has identified a number of putative risk factors of osteoporosis among older adults. Our study suggests that those aged 75–85 years old, of Chinese ethnicity, females, were never married or widowed and with lower education may be at a higher risk of osteoporosis and this group could be targeted for prevention as well as screening and early diagnosis using other diagnostic modalities such as bone marrow scans. Future research should include conducting studies to measure BMD among those with and without risk of osteoporosis based on OSTA cut off score. This would provide a better estimate on the prevalence of osteoporosis. A deeper understanding of the association between risk of osteoporosis and diabetes and hypertension is also needed. It is possible that lifestyle and dietary practices may confer some benefits that can be encouraged amongst all older adults. In conclusion, this study is useful in estimating the allocation of resources and planning of strategies which could potentially prevent osteoporosis and fractures among older adults in Asian populations.

## Figures and Tables

**Table 1 ijerph-16-02338-t001:** Sociodemographic characteristics of the sample by OSTA category.

	Overall Sample		OSTA > −1		OSTA ≤ −1	
	*n*	Weighted %	Weighted %	S.E.	Weighted %	S.E.
Overall	2345	100	48.0	1.37	52.0	1.37
Age group						
60–74	1445	77.0	60.2	1.75	39.78	1.75
75–84	613	18.8	8.6	1.44	91.45	1.44
85+	287	4.2	0.3	0.17	99.66	0.17
Gender						
Males	1052	44.7	62.9	1.92	37.14	1.92
Females	1293	55.3	36.0	1.88	63.99	1.88
Ethnicity						
Chinese	935	83.6	45.7	1.62	54.32	1.62
Malay	651	9.0	59.6	1.85	40.37	1.85
Indian	723	5.9	58.7	1.67	41.25	1.67
Others	36	1.5	64.9	6.26	35.10	6.26
Marital status						
Married/cohabiting	1411	65.0	56.8	1.76	59.26	5.47
Never married	128	8.0	40.7	5.47	43.24	1.76
Widowed	703	21.3	20.4	2.37	79.62	2.37
Divorced/separated	101	5.6	63.3	6.33	36.67	6.33
Education						
None	424	15.7	22.3	3.05	77.65	3.05
Some, but did not complete primary	569	24.1	42.9	2.91	57.06	2.91
Completed primary	604	24.9	50.7	2.92	49.28	2.92
Completed secondary	492	22.7	58.2	3.09	41.81	3.09
Completed tertiary	249	12.7	65.7	4.03	34.31	4.03
Employment status						
Paid work (part-time and full-time)	684	35.3	66.0	2.46	34.02	2.46
Unemployed	31	1.6	59.9	11.80	40.08	11.80
Homemaker	714	26.0	37.6	2.71	62.42	2.71
Retired	895	37.1	37.9	2.26	62.13	2.26

OSTA: The Osteoporosis Self-Assessment Tool for Asians; S.E.: Standard Error.

**Table 2 ijerph-16-02338-t002:** Sociodemographic correlates to OSTA ≤ −1 (at risk of osteoporosis).

	OR+	95% CI		*p* Value
		Lower	Upper	
Age group				
60–74	^ Ref			
75–84	15.6	9.923	24.48	**<0.001**
85+				
Gender				
Male	Ref			
Female	3.5	2.406	5.112	**<0.001**
Ethnicity				
Chinese	Ref			
Malay	0.4	0.334	0.605	**<0.001**
Indian	0.5	0.382	0.666	**<0.001**
Others	0.4	0.145	0.856	**0.021**
Marital status				
Married/cohabiting	Ref			
Never married	2.2	1.265	3.835	**0.005**
Widowed	1.9	1.223	2.902	**0.004**
Divorced/separated	0.8	0.386	1.459	0.397
Education				
None	2.8	1.507	5.203	**0.001**
Some, but did not complete primary	1.8	1.047	2.959	**0.033**
Completed primary	1.8	1.084	2.958	**0.023**
Completed secondary	1.4	0.864	2.338	0.167
Completed tertiary	Ref			
Employment status				
Paid work (part-time and full-time)	Ref			
Unemployed	1.1	0.253	4.745	0.902
Homemaker	0.9	0.579	1.433	0.686
Retired	1.7	1.175	2.364	**0.004**

Bold font indicates significant *p* values; OR+: +Odds Ratios derived from multiple logistic regression using stepwise method; ^ Ref: Reference.

**Table 3 ijerph-16-02338-t003:** Clinical conditions correlates to OSTA ≤ 1 (at risk of osteoporosis).

	OR+	95% CI		*p* Value
		Lower	Upper	
Hypertension				
No	^ Ref			
Yes	0.6	0.4	0.9	**0.008**
Heart Problems **				
No	Ref			
Yes	0.9	0.6	1.3	0.514
Diabetes				
No	Ref			
Yes	0.6	0.4	0.9	**0.007**
TIAs				
No	Ref			
Yes	0.6	0.2	2.3	0.491
Stroke				
No	Ref			
Yes	0.9	0.5	1.7	0.859
10/66 Dementia				
No	Ref			
Yes	0.9	0.4	2	0.744
Depression ***				
No	Ref			
Yes	0.9	0.6	1.3	0.571

Bold font indicates significant *p* values; OR+: +Odds Ratios derived from multiple logistic regression adjusted for sociodemographic variables; ^ Ref: Reference; TIA: Transient ischaemic attack; ** Heart problems (heart attack, angina, heart failure, valve disease & others); *** Depression (lifetime diagnosis based on respondent/informant self-report).

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
