# Peer review of "Estimation of Prevalence of Osteoporosis Using OSTA and Its Correlation with Sociodemographic Factors, Disability and Comorbidities"

_ijerph, 2019, doi:10.3390/ijerph16132338_

Round 1

Reviewer 1 Report

I have reviewed the revised version and no further comments to be addressed.

Author Response

Thank you for reviewing the paper and your comments. We have checked through the paper for the language and have made the necessary corrections and amendments.

Reviewer 2 Report

I think that the purpose of the research is clearly stated and greatly improved by the author's current revision.

Document 14 is the first publication of OSTA. The authors state that OSTA is designed for both men and women, but reference 14 states that it was designed for women. Moreover, in the reference 14, it is described that all subjects are women.Therefore, OSTA is originally designed for women. Reference 23 states that it has been confirmed that it is possible to screen men with low bone density using age and weight as well as OSTA. Therefore, in the introduction chapter, it should be stated that OSTA was originally designed for screening Asian women, but it was also reported to have been effective for men using different cut-off value.

There is no mention in the title that socio-degraphic information was also considered in this study.

In addition, although the title describes the correlation with disability and co-morbidities, there are few descriptions in the discussion and conclusion chapters. I recommend you to improve the description of this content.

Page3 Line19 [24] ? [23]?

Author Response

Thank you for your suggestions. As suggested, we have revised the introduction to include the use of OSTA for males, page 2, line 18 and 19. As advised, we have also changed the title of the paper to include the use of socio-demographic factors for the paper.

Thank you for your comments regarding the discussion and conclusion section. We have explained the correlations of co-morbidities in page 7, line 33 to 48. For correlations to disability, this is explained in page 7, line 49 to page 8, line 2. As per your advice, we have further added in more description in the discussion section, page 7, line 39-40. We have also amended our conclusion section, to reflect correlation of socio-demographic factors, disability and co-morbidities.

Separately, we apologise for the error in the referencing, we have amended it accordingly.

This manuscript is a resubmission of an earlier submission. The following is a list of the peer review reports and author responses from that submission.

Round 1

Reviewer 1 Report

I have several questions as to the study:  

#1. The title of this article depicts "Estimation of Prevalence of Osteoporosis..." However, you just mentioned about OSTA in your study. OSTA score does not represent the actual entity of osteoporosis. The title might warrant revision.  

#2. The reason that disability assessment didn't reveal significant finding might be caused by inappropriate comparison. Osteoporosis is regarded as a kind of silent epidemic.  Lots of people in osteopenia might not be aware of his/or her problem of osteoporosis, not to mention the phenotype disability.   

#3. The cut-off value of -1 (OSTA) harbors different specificity and sensitivity between female and male. You also pointed out that female is at a higher risk of osteoporosis (OR:15.6). Since lots of confounders related to gender differences(ex. estrogen level), it's inadequate to generate conclusion based on the analysis of all cohort. Why not focus on female cases merely? 

#4. The r square of the regression analysis is needed for providing more convincing statements. 

Reviewer 2 Report

This is a cross sectional survey to estimate the number of older adults at high risk of osteoporosis using OSTA. It is overall well written, although some typos and grammatical errors need to be fixed. While BMD was not available for study subjects and screening tool OSTA was used instead. Study subjects are primarily elderly >60 in Singapore. I am having difficulty finding the clinical value of this paper since osteoporosis prevalence was not directly measured. I would suggest authors focus on clinical value of the study findings in discussion. For instance, risk factors detected in this study were all non-modifiable factors (age, female, non-Chinese, never married or widowed, lower education and retired), two protective factors detected (HTN and DM) likely associated with metabolic syndrome and high BMI. How can the study findings be used to help improve bone health in Singapore residents. For those subjects labeled as high risk for osteoporosis, what have been down by the research team (ie. referring for DEXA scan, starting prevention or treatment). This is important to disclose to show the clinical value of this study.    

Reviewer 3 Report

Overall comment

OSTA is being developed for the screening of osteoporosis in Asian women.However, in this paper men are included in the subjects. I think this is a fatal flaw in this research.You have to justify including men in the subject of this study.Or men should be excluded from this study. Also, as osteoporosis is often asymptomatic until a fracture occurs, being aware of physical disability may be difficult unless the patient has a fracture.Therefore, I think that the lack of relevance between the OSTA score and the physical disability, which is the purpose of this research, was predictable.

Individual comment

Page2 Line26: I recommend changing "establish " to "estimate"  according to the title of the paper.

Page3 Line19: "...using a 12 item" You should explain these 12 items specifically.

I think that there is a lot of content different from the consideration from the results of this study in the discussion chapter.

For example, Page6 line26 - Page7 Line2, "In 1992, ...elsewhere in the world" 

Like these contents may be better described in the introduction.